# Workers’ Health under Algorithmic Management: Emerging Findings and Urgent Research Questions

**DOI:** 10.3390/ijerph20021239

**Published:** 2023-01-10

**Authors:** Emilia F. Vignola, Sherry Baron, Elizabeth Abreu Plasencia, Mustafa Hussein, Nevin Cohen

**Affiliations:** 1Department of Community Health and Social Sciences, City University of New York Graduate School of Public Health and Health Policy, 55 West 125th Street, New York, NY 10027, USA; 2Barry Commoner Center for Health and the Environment, Queens College, City University of New York, 311 Remsen Hall, 65-30 Kissena Blvd, Queens, NY 11367, USA; 3Department of Health Policy and Management, City University of New York Graduate School of Public Health and Health Policy, 55 West 125th Street, New York, NY 10027, USA

**Keywords:** algorithmic management, platform work, gig economy, worker health, work stress

## Abstract

Algorithms are increasingly used instead of humans to perform core management functions, yet public health research on the implications of this phenomenon for worker health and well-being has not kept pace with these changing work arrangements. Algorithmic management has the potential to influence several dimensions of job quality with known links to worker health, including workload, income security, task significance, schedule stability, socioemotional rewards, interpersonal relations, decision authority, and organizational trust. To describe the ways algorithmic management may influence workers’ health, this review summarizes available literature from public health, sociology, management science, and human-computer interaction studies, highlighting the dimensions of job quality associated with work stress and occupational safety. We focus on the example of work for platform-based food and grocery delivery companies; these businesses are growing rapidly worldwide and their effects on workers and policies to address those effects have received significant attention. We conclude with a discussion of research challenges and needs, with the goal of understanding and addressing the effects of this increasingly used technology on worker health and health equity.

## 1. Introduction

Discussions about the changing nature of work have focused on the rapid expansion in the commercial use of algorithms, mathematical formulas that make autonomous decisions based on procedural rules or statistical models [1,2], to accomplish core management functions in a wide range of industry sectors. Perhaps the most visible and extreme example of this new form of management is its application in digital labor platforms, companies that organize and manage transactions among buyers, sellers, and workers, through digital technologies like websites and apps. While algorithmic management replicates many features of labor control strategies under human management, its specific use in digital labor platforms, where algorithms virtually replace human supervisors in employee performance monitoring, scheduling, compensation, and hiring and termination, is a relatively recent phenomenon that is experienced by a growing number of workers globally [3]. Existing surveillance systems are not well equipped to measure the exact size of this workforce in part because of a lack of clear, measurable criteria for classifying workers as part of the gig economy. The International Labor Organization has described gig economy work as including “crowdwork”, which entails completing short tasks (e.g., identifying the content of a photo) from anywhere in the world through online platforms such as Amazon Mechanical Turk, and “work on demand via apps”, which involves traditional working activities (e.g., transport, cleaning and running errands, delivery services), performed locally through platforms such as Uber/Lyft, Doordash, and Taskrabbit [4]. For the most part, online platform companies view gig workers as independent contractors with at least some ability to select or refuse jobs, and set their hours and the level of participation. However, platform companies also control to varying degrees many aspects of the work, for example by determining who can provide their services, how tasks are assigned, and the pay rate for tasks [5]. In the US, a 2021 Pew Research Center survey, which used a definition of gig work more consistent with “work on demand via apps,” found that 16% of US adults had made money at some point through an online “gig” platform, with 9% reporting that they had earned money this way in the previous year. Of those current or recent gig workers, 31% (3% of all US adults) reported this work as their main job [6].

Scholars have recently begun to investigate, through largely qualitative methods, the implications of these technological changes for worker health and well-being [7,8]. This research suggests that algorithmic management likely influences several dimensions of job quality with known links to worker health, including workload, income security, task significance, schedule stability, socioemotional rewards, interpersonal relations, decision authority, and organizational trust (see Figure 1). A recent review examining the functions and consequences of algorithmic management concluded that workers’ negative reactions to algorithmic control likely exceed their positive reactions, especially as related to the workers’ sense of autonomy, the power imbalances created by information asymmetry, and the perceived opacity and unfairness of algorithmic decision-making [9]. However, researchers argue that negative outcomes are not inevitable; rather, algorithms can be designed and implemented in ways that might balance organizational needs with worker needs, especially if done with worker input [9,10]. Understanding the effects of algorithmic management on worker well-being is essential to that process.

Platform-based food delivery work, both for groceries and prepared meals, offers a specific but generalizable example with which to explore how algorithmic management can influence job quality and health. Though aspects of algorithmic management have been adopted throughout diverse industry sectors, its application in grocery e-commerce and restaurant food delivery has accelerated rapidly, especially since COVID-19, as lockdowns and social distancing recommendations curtailed in-person shopping and increased eating at home. Today, hundreds of thousands of workers are employed by digital platform companies as independent contractors delivering meals and groceries. They are disproportionately low-wage, immigrant, and people of color. Because these demographic groups are overrepresented in such newer forms of work that rely on algorithmic management, understanding how this trend shapes their work and health is critical to addressing employment-related health inequities. Furthermore, local governments are beginning to regulate food delivery work, and the US Department of Labor has proposed rules that could reclassify many gig workers as employees [11], making health research on platform-based food delivery workers particularly timely.

In this article, our mission is twofold: first, we synthesize emerging conceptual and empirical findings on workers’ health under algorithmic management from interdisciplinary literature in public health, sociology, management science, and human-computer interaction studies, focusing on platform-based food and grocery delivery workers; second, we identify research priorities and offer guidance for health researchers. We begin with an overview of the platform-based food and grocery delivery workforce, followed by a description of the features and mechanics of algorithmic management in that work. We then synthesize existing research on the ways algorithmic management influences key dimensions of job quality and its documented health impacts. The occupational health literature on this topic is limited, especially empirical studies examining health outcomes associated with algorithmic management processes. Thus, rather than a structured review of studies, we summarize current literature exploring how algorithmic management is changing the work process and the magnitude and types of stressors experienced by workers, specifically those in food delivery jobs. Based on the gaps in this literature, we conclude with a discussion of research challenges and priorities to better understand and address the effects of this increasingly prevalent technology on worker health and health equity.

## 2. Platform-Based Food and Grocery Delivery Workers

The market for food delivery services, recently estimated to be worth USD 150 billion worldwide, has more than tripled since 2017 [12]. In the US, food delivery sales more than doubled during the COVID-19 pandemic as consumers avoided grocers, restaurants limited indoor dining, and people spent more time at home [12,13]. The increased demand for food delivery has been especially enabled and accompanied by the emergence of digital platform-based food delivery companies. Platform-based food delivery work is now relatively common: a 2021 Pew Research Center survey found that 7% of US adults reported delivering meals and 4% reported shopping for or delivering groceries through an app, compared to 5% who reported driving for a ride hailing app [6]. While some do this work part-time to supplement another job or accommodate childcare or other family responsibilities, data collected in New York City, San Francisco, and across multiple US states suggests a substantial portion (50% or more of workers depending on the source) engages in this work full-time [14,15,16,17,18]. In some cities, the size of the food delivery workforce is large; for example, in 2021, New York City had approximately 65,000 app-based prepared food delivery workers [17]. The gender distributions of these workforces appear to differ in terms of type of food delivery and location. In New York City, where over 50% of prepared food delivery workers rely on non-car modes of transportation, data from 2021 indicate the worker population is predominantly male [18]. Among grocery e-commerce work, where car deliveries predominate, studies conducted between 2018 and 2021 in multiple parts of the US found that women made up at least half of the workforce [15,19].

Across the US, gig work in general, and platform-based food delivery work in particular, attracts young, low-income, and Hispanic workers [6]. This is largely because the work has low barriers to entry and thus is easily accessible to those with limited previous work experience, recognized credentials, or fluency in the local language. Few platforms ask workers to provide evidence of their qualifications or skills when setting up their profile [20]. Workers need only smartphones and access to bicycles or vehicles to do food delivery work, and app messaging makes speaking the local language fluently less important for getting and carrying out delivery jobs [21]. Platforms have quick onboarding practices that enable those urgently in need of income to begin working quickly [22]. Some have argued that the low barriers to entry enable immigrants, who may have difficulty finding other work because of language or credentialing requirements, to earn money, while others suggest that demand is the result of racialized laws and policies that exclude immigrants from other types of work and therefore make them more likely to seek out platform-based work [22].

## 3. Features of Algorithmic Management in Platform-Based Food Delivery

Platform-based food delivery work falls within the broader category of “geographically tethered” work on demand via apps, which requires workers to be in a specific place to complete the work, as opposed to “cloudwork” that people can do from anywhere with a computer and internet connection [23]. In platform-based food delivery, all management functions, from the assignment of jobs, the tracking of hours, pay (including tips), and performance evaluation, are conducted through the company’s app, the mobile user interface of the algorithm [24]. For example, when a customer places a restaurant food order on an app, workers who are signed into the app are assigned a delivery based on their location in relation to the restaurant and customer. The app directs the worker to the restaurant for pick-up, and then to the customer address for delivery. Generally, the only contact a worker has with the platform company is to accept or reject the job, and interaction with the customer is limited to handing over the delivery, though customers can voluntarily include a tip on the app or in cash at the door. For supermarket delivery, customers complete orders through a website or mobile app and then workers at specific grocery stores are directed by the app to fulfill the order. The app provides efficient shopping patterns based on product location on the shelves. For some grocers, shoppers work in dedicated order fulfillment facilities where they are guided by other technology to improve the efficiency of selecting and assembling the products. Grocery customers either pick up their order at the store or choose to have paid workers deliver it to their homes. In some cases the same worker both shops and delivers, and in other cases different workers perform these separate job functions. Some grocery stores employ the in-house shoppers as regular employees rather than as independent contractors [15]. 

The algorithmic management seen in platform-based food and grocery delivery work replicates many features of past strategies to control labor [25], from Taylor’s scientific management of factory work to the technological surveillance, monitoring, and speed-up of service sector work [23]. In platform work, however, the precise timing and measurement of the labor process happens outside a physical workplace, with workers’ actions meticulously tracked through their mobile devices and other software, and it is the algorithm, rather than a human supervisor, that makes decisions based on the data. These decisions determine work schedules, speed, compensation, and termination. Data are collected on the rates of task acceptance, rejection, and completion; the locations of pick-up and delivery; the distance and speed of travel; and customer ratings and tips in a form of digital surveillance that is constant yet opaque to workers, with companies citing the proprietary nature of the algorithms [2,10,21]. Furthermore, continuous monitoring allows predictions about workers’ future behaviors, which are turned into operational decisions such as work scheduling or “fitness for employment” [26]. Platform companies design algorithms to use gamified bonuses, rewards, and other incentives to encourage workers to internalize management’s aims, stay logged into the apps, and accept difficult jobs during periods of peak customer demand. The efficacy of automated supervision and management relies on “the social power of algorithms” [27] (p. 22), which includes detailed supervision in the form of electronic messages to workers and enforcement of discipline through positive signals (such as compensation bonuses) or negative signals (such as deactivation from the platform), all without human intervention and with limited or no opportunities for workers to obtain feedback or contest decisions.

## 4. Dimensions of Job Quality Affected by Algorithmic Management

Building on several recent reviews [7,9,28], this section summarizes literature relevant to the following research question: how do features of algorithmic management influence the specific dimensions of job quality, including both work and employment quality [29], that have been associated with worker health and well-being? We began with established job stress models (Job-Demand-Control-Support [30] and Effort-Reward Imbalance [31]), along with more recent research on employment quality [32,33], to guide our choice of specific job quality dimensions. As we reviewed the available literature, we grouped findings by dimension in an iterative process, re-adjusting the categories until we felt they synthesized the findings adequately and comprehensively. We summarize the dimensions of job quality that may link algorithmic management to worker health in Figure 1. These job quality dimensions are not mutually exclusive, and algorithmic design can have synergistic effects on multiple dimensions. However, for conceptual clarity, we discuss each job quality dimension separately. 

Though much of the research synthesized below points to negative effects of algorithmic management on job quality, in Figure 1 we list the functions of algorithmic management and dimensions of job quality in neutral terms to suggest this is not inevitable. Rather, as Zhang et al. [10] and Parent-Rocheleau and Parker [9] argue, algorithms can be designed and implemented in ways that balance the platforms’ business needs with workers’ needs, particularly if the considerable power imbalance between workers and platform companies that exists in most current uses of algorithmic management were addressed, whether voluntarily by the companies, through worker organizing, or through regulations. As the diagram shows, the health impacts of algorithmic management are also shaped by various contextual factors, including developments in the technology used by the platform companies, demand for the services these companies provide, labor and social welfare policies and legislation, and the broader social contexts in which workers live. These also represent potential intervention points that can modify the health effects of algorithmic management on workers once those effects are better understood.

### 4.1. Workload

Algorithmic management influences workload, both the amount and pace of work tasks. Delivery workers may adopt risky behaviors such as speeding, using mobile phones while driving or cycling, running red lights, and skipping lunch and bathroom breaks to meet customers’ expectations of fast deliveries, increase earnings by completing more orders, or avoid fines imposed by platform companies in case of delays [34]. The effects of work pressure may vary by length of time on the job: in one interview-based study, newer workers felt more pressure to deliver quickly than longstanding couriers who learned to pace themselves [35]. In inclement weather, when more consumers order food for home delivery, the increased demand exposes delivery workers to extreme weather and may increase accident risks, especially with bicycle delivery [36]. A study of platform delivery workers in India found that they were motivated by a higher pay per task to deliver food at night despite more dangerous situations where they could be assaulted or robbed [37]. A survey of Uber Eats workers in Tokyo during the pandemic found that economic precarity encouraged more of them to work even under unsafe conditions, such as heavy rain or fatigue [38].

### 4.2. Income Insecurity

Piece rate pay, dynamic pricing, the lack of transparency about pricing, and one-sided changes in both pay rate and in the way tips are handled makes it difficult for workers to predict their income [14,39]. Platforms use algorithms to track metrics (such as acceptance rates, hours worked, and customer ratings), to rank workers, and to give those with higher rankings a chance to choose orders first or turn down orders without penalty, affecting workers’ potential earnings [39]. Workers who testified during a 2022 hearing of the New York Department of Consumer and Worker Protection described the challenges of income insecurity and instability that come with working for delivery platforms [40]. The department subsequently reported survey data indicating that app-based restaurant delivery workers in New York City earn USD 4.03 per hour without tips and USD 11.12 per hour with tips, after hourly expenses are accounted for [18]. At the hearing, across workers’ testimony there were clear patterns of discontent regarding the lack of transparency, which makes it difficult for them to understand what they can do to earn more while prioritizing their desire to have greater control over their work [41]. Workers cited the varying demand for app-based deliveries throughout the year as the reason why delivery work can change from a reliable source of income to a source of uncertainty, suddenly requiring more hours on the apps and in the streets to cover one’s living expenses. Furthermore, workers pointed out that incentives promised by Uber Eats and DoorDash to bring in new workers are not long lasting. Many expressed that their first months working for the platforms had been profitable, while requiring fewer hours a day, but that after this period their new normal had become taking on daily 12-h shifts to reach the same net income [41]. 

### 4.3. Task Significance

Algorithmic management influences task significance, or the extent to which a worker considers a job important, or believes it to influence others’ lives [42]. The way algorithms handle compensation, rewarding quantified and decontextualized activities, (e.g., the number and speed of deliveries regardless of broader conditions such as weather, delays with orders out of workers’ control, or problematic customers), can diminish task significance [43]. This type of monitoring and compensation can push workers to orient their energy towards the quantified aspects of a job, which might not be the most meaningful or complex ones, a process some call “datafication”, or working for data [44]. Datafication can lead to alienation, detachment from work, and decreased creative thinking [28]. Further, algorithms also limit the ability of workers to engage in work customization (e.g., devising a delivery route), which can make work feel less meaningful [45]. Gamified incentives to take on additional jobs during peak periods, or to remain on the platform to complete additional jobs, can turn repetitive tasks into challenges that enable workers opportunities to “win” by hitting bonus targets. Rather than encouraging creative problem solving, however, gamification likely feeds into datafication and its competitive nature may increase stress by orienting workers’ energy towards these quantifiable “wins” [46].

### 4.4. Schedule Stability

Just-in-time scheduling to match labor supply to consumer demand for services in real time saves companies and customers labor costs in the short term, because workers are only paid for the moments they are actively completing a task, such as a delivery, but it shifts these costs to workers in the form of unstable schedules and uncompensated time awaiting the next gig [47]. Some platform companies (e.g., DoorDash, Instacart, GrubHub, Relay) allow or require workers to choose shifts in advance, but they all put the onus on workers to find the next job to complete, in contrast to conventional employment, in which owners or managers are responsible for securing business [48]. The need to work consistently throughout the day and to accept jobs as they become available exacerbates stress among platform workers. A survey of ride share and food delivery platform workers showed that they usually had to work through peak hours, typically lunch and dinner time, with two-thirds saying they felt stressed about taking longer breaks during their peak work hours [49]. Insecurity may also pressure workers to accept jobs when pay is highest, such as late at night or in inclement weather, when the work is not only more physically challenging and dangerous but mentally stressful as well [50]. Unpredictable hours may blur the boundaries between work and personal times and spaces, preventing workers from achieving a healthy work-life balance [20]. 

### 4.5. Socioemotional Reward

Algorithms rely heavily on customer feedback in the form of rankings or tip amounts to allocate platform work and potentially remove workers deemed inadequate [51]. The dependence on good customer ratings to remain on a platform and the need to earn tips also pressures delivery workers to perform emotional labor to please customers, which can be mentally exhausting [35,52]. To improve their reputation, platform workers may avoid taking breaks or cancelling work for any reason, including illness. To avoid the threat of reduced work or deactivation from a platform, many app-based workers spend time outside of their platform work disputing negative ratings with company representatives, or “reputation auditing” [50]. Limited means to explain poor customer evaluations to an app, compared to a human supervisor or manager, means platform workers cannot reverse poor ratings that may be based on situational factors beyond their control (e.g., poorly prepared food) or the racial, ethnic, or gender biases of customers. 

### 4.6. Interpersonal Rewards

The absence of interactions with a manager can be health-promoting for some, but it also means no person to provide support. In effect, interpersonal relations with both managers and other workers are drastically altered in platform-based work. The mode of communication between the platform company and workers via an app is a barrier to explanations for decisions, which can lead workers to believe they are being ignored, mistrusted, treated unfairly, or insufficiently valued [35,45,53]. Platform workers are often physically separated from each other and from their supervisors, and the absence of opportunities to engage directly and dialogically through the apps with supervisors or other workers to exchange information, engage in formal or informal training activities, and build a shared safety culture can lead workers to feel isolated [20,54]. The apps can encourage competition among workers to “win” desirable jobs, which reduces cooperation and amplifies the sense of isolation [55]. Feeling isolated can spill into life outside work. Interviews among platform-based food delivery workers in India, for example, found that because these workers were not affiliated with a specific restaurant and spent most of their time making deliveries, they had few opportunities to see friends and family and therefore experienced loneliness [37].

### 4.7. Decision Authority

Flexible work schedules typical of platform work may be experienced by some workers as a source of autonomy and may contribute to well-being and work-life balance by enabling them to work when and where it fits their life [20]. However, for those financially dependent on consistent work, the flexibility also involves atypical working hours and the need to do multiple jobs to earn the equivalent of a conventional wage, both of which add to stress [35,50,56]. Information asymmetry also limits access to useful information that could help workers make informed, beneficial decisions, effectively limiting their choices regardless of their perceived schedule flexibility. Platform companies’ unliteral changes in how work is organized, compensated, or evaluated, often without explanation to workers, contribute to a perception of the algorithm as an arbitrary authority, with uncertain implications for workers’ pay and ability to plan their work and lives [14,57]. 

The metrics that platforms collect from and about workers (e.g., their location and speed), gamification, and publication of work statistics are mechanisms of “techno-normative control” that lead to altered behaviors that deviate from the choices platform workers would make if they were working within a system that allowed truly autonomous decisions [14]. For example, the system monitors the rate at which workers accept orders and punishes workers who repeatedly decline work, including if they decline an order due to safety concerns [58]. Requiring workers to communicate with the platform company via the app is also a barrier to providing explanations for decisions and therefore is a mechanism to limit workers’ ability to respond to problems [35,45,53]. 

Despite the considerable limits to worker decision authority, platform workers engage in strategies to game the algorithms and resist the rules embedded in them. Many platform delivery workers sign up with multiple companies and alternate among different platforms. Their decision to work with a particular platform depends on which company provides the best pay and bonus incentives on a given day or week [2,46]. Other strategies include declining certain jobs in anticipation of more lucrative gigs, signing out from a platform when undesirable jobs are likely to be offered, or anticipating surge pricing. However, gaming or resisting the algorithms requires knowledge, know-how, and risk tolerance, making it easier for some workers and harder for others. 

### 4.8. Organizational Trust

Researchers have also investigated the influence of algorithmic management on workers’ perceptions of fairness and levels of organizational trust, or workers’ confidence that an organization will perform in ways that are not harmful to them. Workers find it difficult or impossible to negotiate or ask for feedback from a platform, for example when trying to reinstate an account if suspended by the app [20]. An awareness of the imbalance in power between themselves and the platforms causes many workers to feel that they are being cheated and exploited, leading them to harbor resentment towards the platforms and the algorithms that manage them [10,57]. Specifically, workers often trust a human in a management position to execute decisions that would be beneficial to both the company and its employees, while they are more cautious of errors and biases integrated into algorithms [43,59].

## 5. Health Impacts of Platform-Based Food Delivery Work

Much of the previously referenced literature aimed to document the major types and sources of work stressors associated with algorithmic management but did not systematically measure health outcomes. The occupational health research that has been conducted on food delivery work has focused mostly on safety issues and injury rates. Platform workers who perform physically strenuous work such as food delivery face occupational hazards similar to those of conventional employees doing the same type of work [54,60]. Food delivery is inherently risky as it involves the possibility of accidents, especially at night or in inclement weather, and the challenges that can arise when entering homes and interacting with different unknown individuals. Bicycle injury reports are common among food delivery workers. According to one estimate, app-based food delivery workers have an injury rate 16 times higher than construction laborers [18]. A recent scoping review found that time-pressured work and inadequate protective gear were common underlying factors of elevated injury rates [61]. Food delivery workers also report challenges to bathroom access [17], which has been reported in taxi drivers and is associated with adverse urological outcomes [62]. During the COVID-19 pandemic, the risk of contracting the virus due to contact with potentially infectious restaurant employees or clients, and the possible financial burden of missing work, has caused mental stress among delivery workers [63].

As platform food delivery workers are classified as independent contractors, they are most often required to provide their own safety equipment, such as helmets [61]. Like other workers in contract, contingent, part time, and other forms of non-standard employment, platform workers face occupational health risks from having to work at different sites, under changing conditions, and with less on-the-job experience than long-term employees, factors that make understanding safety precautions and methods of risk mitigation more difficult, increasing the likelihood of on-the-job injury. Employers of platform workers may also be less willing to invest in training, protective equipment, and close supervision [64].

In addition to safety and health risks associated with delivery work in general, algorithmic management tools can create significant job-related psychosocial stressors and may increase the likelihood of collisions, skipped bathroom breaks, and other work risks. For example, pressure to increase earnings and app-related pressures to make delivery targets can cause increased work pace and long work hours. This, in turn, can cause workers to cut corners, forego basic health and safety protections, and increase fatigue and other risk factors for work-related injuries [54]. For example, a survey of platform delivery workers in Ho Chi Minh City during the COVID-19 pandemic found that longer work hours, a larger geographic area to cover, and pressure to perform led riders to take fewer precautions, such as sanitizing their hands, wearing masks, and riding carefully [36]. The frequency with which algorithms offer new orders and require workers to respond creates stress [65]. Financial pressure to overwork or to work irregular hours may lead to perceptions of insecurity, which are associated with anxiety, anger, depressive symptoms, poor sleep, and poor self-rated health [50,66,67]. Worry about the platform rating system and its effect on the jobs workers are assigned and their tips are positively associated with stress, poor sleep, and depressive symptoms [50]. Finally, the absence of opportunities to interact, share knowledge, or build a shared safety culture through the apps with supervisors or other workers can lead to isolation, stress, and negative mental health [20].

## 6. Research Needs

The specific health and well-being impacts of algorithmic management are likely shaped by the role of algorithms in the work overall, the nature of the work, and characteristics of the workers doing it. However, the dimensions of job quality described in this article are likely influenced in some way by algorithmic management among all workers who experience this increasingly prevalent technology. New and expanded research is needed to better understand the role of algorithmic management in worker health, including improved occupational injury and illness surveillance data, as well as expanded research on pathways between platform work and health and on its implications for health equity.

### 6.1. Surveillance

Platform workers, including app-based food delivery workers, are not adequately captured in existing surveillance systems. Their conditions of employment are sometimes tracked in broader public health surveillance systems along with other social determinants of health, but these systems seldom collect employment data at sufficient detail to identify platform workers in specific industries. The US Bureau of Labor Statistics’ Contingent Worker Supplement of the Current Population Survey collected some of this data, but is conducted infrequently and has not been designed to collect data related to platform employment specifically [68]. As independent contractors, gig workers are not counted systematically by the Occupational Safety and Health Administration or in workers’ compensation injury and illness statistics. Moreover, even for workers who are included in these systems, occupational safety and health impacts other than work-related traumatic injuries are largely underreported [69]. Creating common definitions and measurement tools to identify platform gig workers will enable the collection of high-quality surveillance data that could help: (1) support more accurate estimates of the size and composition of the workforce; (2) identify disparate patterns of adverse health outcomes potentially tied to work; and (3) monitor changes over time in both the number of these workers and health outcomes, especially when new intervention policies and programs are implemented. 

### 6.2. Pathways between Platform Work and Health and Effects on Health Equity

Because algorithmic management shapes how platform work is organized, more theory-informed research is needed to better understand the pathways through which algorithmic management might affect worker health and well-being. The limited available empirical evidence linking algorithmically influenced dimensions of job quality directly and indirectly with workers’ health and well-being suggests directions for additional research, especially studies of pathophysiologic changes and subclinical disease. For example, safety science literature suggests that safety culture emerges among workers who are able to interact and communicate with one another because these connections foster shared values, norms, and perceptions of good work practices [54]. In contrast, algorithmic management fosters dispersion of the workforce, which may hinder the development of safety culture and affect injury rates. Customer rating systems that are a core feature of algorithmic management pressure workers to perform emotional labor such as “surface acting” (i.e., pretending to care), which has been linked to impaired well-being, emotional exhaustion, work-family conflict, negative job attitudes, insomnia, and psychosomatic complaints [35]. We do not yet have a clear understanding of the burdens of these health outcomes among workers who experience algorithmic management. Perceived fairness and organizational trust have implications for workers’ job satisfaction, emotional well-being, and physical health, yet we do not know the extent to which feelings of unfairness and untrustworthiness contribute to these health issues among platform workers [70]. Quantified monitoring and evaluation can lead to work alienation, detachment from work, and finding work meaningless, which have been associated with decreased well-being, job dissatisfaction, and increased emotional exhaustion [35]. Understanding the degree to which these associations exist among platform workers is critical to ameliorating these conditions. 

In recent decades, several research groups have developed frameworks that guide research designs and methods for better assessing how employment quality and working conditions impact the health and well-being of those who engage in non-standard and precarious work [32,33,71], such as food delivery platform work. While the focus of this article is on the potential health effects of algorithmic management specifically, as is shown in Figure 1, these health effects are likely moderated by other work conditions, such as the physical demands of a job, and dimensions of employment quality, such as employment stability associated with a particular employment arrangement. The way these underlying employment characteristics intersect with algorithmic management to influence worker well-being is a relationship public health researchers and practitioners must understand better.

The family and community impacts of platform work in general, and algorithmic management specifically, as sources of work stress should also be included in a broad research agenda. Unpredictable work shifts, extended work hours, income insecurity, and responsibility for maintaining essential work equipment likely create stressors for workers’ extended families. These stressors affect whole communities by compounding problems caused by inadequate housing quality, food insecurity, limited household savings, poor educational options for children, limited health care access, and constrained eligibility for supportive entitlements. Analyzing the ways that work stressors influence community health will contribute to a more comprehensive understanding of the social costs of the current systems used to sustain digital platform labor.

Finally, research on the health effects of algorithmic management should be formulated with a focus on equity. There are four elements to unmasking inequities in algorithmic management. First, the algorithms themselves may create inequity. Predictive models may be designed to make less biased management decisions than humans. On the other hand, they may use performance or behavioral response data that is itself biased or may be programmed in a way that perpetuates or amplifies the biases of developers, thus exacerbating racial, ethnic, gender, and other disparities while hiding the sources of such disparities within the algorithms, making it harder to detect and address inequities. Second, the process of interacting through algorithmically mediated devices may lead to inequitable outcomes. In addition to biases embedded in the algorithms themselves, dependence on specific technology (i.e., apps, phones) to interact with one’s employer, the human and spatial isolation enabled by the algorithm, the type of communication required by the apps (i.e., texting, chat bots), and the difficulty with which workers can interact with humans to solve problems may disadvantage those with limited language or communication skills, low confidence, limited social connections, and other socio-emotional characteristics that may differentially affect people based on physical or cognitive abilities, or race/ethnicity, class, and gender. Third, the nature of these jobs may be more harmful to some groups of workers. Platform companies argue that algorithmic management enables workers to function autonomously as individual entrepreneurs (although deviations from expected performance lead to negative consequences for workers) and that independence is a positive feature of platform-based work. We know very little, however, about how different groups of workers, with diverse social and economic needs, experience the benefits or constraints of the specific forms of autonomy and individuality offered by app-based employment. Fourth, it is important to understand the differential effects of algorithmically managed work in the context of the larger economy. App-based gig work exists in large part because many workers are excluded from better quality jobs due to discrimination and policies that dampen wages, inhibit union organizing, and misclassify gig workers as independent contractors. The specific economic context could explain whether, to what extent, and why algorithmically managed work may lead to stigma and other factors that make the experience of these jobs less satisfying and more emotionally fraught.

## 7. Limitations

This article has three important limitations: (1) it synthesizes published research and identifies research gaps, but as a focused (i.e., non-systematic) review, it may be subject to publication and selection bias; (2) most studies cited used qualitative methods, often of small convenience samples or single cases, generating rich theoretical explanations but results that are not statistically generalizable to the larger population of algorithmically managed workers; and (3) the article does not consider the impacts of algorithmic management on non-food platform-based workers or non-platform workers.

## 8. Conclusions

Because of its influence on various dimensions of job quality, algorithmic management likely influences worker health and well-being. Researchers have only begun to scratch the surface of the health effects of algorithmic management and need to conduct more empirical studies examining the short- and longer-term health effects of this type of work. Importantly, this form of management does not necessarily preclude the possibility for worker agency, resistance, and collective organization [27], themselves drivers of job quality and therefore worker health. Furthermore, algorithms can be designed and implemented in ways that balance organizational needs with workers’ needs [9,10]. Similarly, policies to change labor and social welfare laws and regulations are potential intervention points that can influence the effects of algorithmic management on workers’ health once those effects are better understood. Future research will benefit from interdisciplinary collaborations that draw on diverse fields such as computer science, occupational safety and health, sociology, labor studies, public policy, and others, and that draw on the leadership of worker organizations, to better understand—and to intervene effectively on—the ways that algorithmic management influences work design and work stressors.

## Figures and Tables

**Figure 1 ijerph-20-01239-f001:**
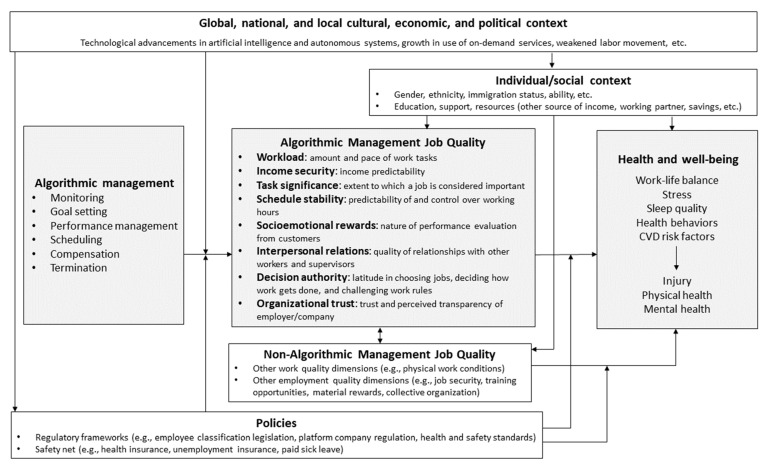
Health-related job quality dimensions influenced by algorithmic management.

## Data Availability

Not applicable.

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
