# Peer review of "Workers’ Health under Algorithmic Management: Emerging Findings and Urgent Research Questions"

_ijerph, 2023, doi:10.3390/ijerph20021239_

Round 1

Reviewer 1 Report

This is a nice synthesis on the subject of platform work, its impacts and challenges and I like the fact that there are proposals for future research. However there are no research results here per se. I don't know the journal enough to know if this is a problem for you but in any case, I do think that this literature review is interesting and useful, and contains many references. there might have been other references to add, but for an article, this is sufficient here.

Reviewer 2 Report

This is a well-written paper, which addresses a timely topic and draws on literature from different disciplines. The following comments are provided to help strengthen the paper. 

Introduction - the changing context of work is generally well-described. It would have been helpful to briefly define gig work and it differs from conventional employment. Second, not all gig work is mediated by digital platforms), and to briefly speak to the global drivers for these changes (e.g. how has globalization and deregulation contributed to these trends in automation and particularly impacted certain industry sectors, and why?)

Good justification provided for focus on platform-based food delivery work, clear description of how this work is conducted, and who performs this work in the US context with implications for employment-related inequities. 

Aims for research are clearly presented. What research questions informed your review? 

Job quality dimensions - are did you arrive at these dimensions? They are reasonable but did you use a framework or were they inductively derived from the literature reviewed?

Figure 1 - very effective presentation 

Section 4: 

Task significance - what do you mean by "quantified and decontextualized activities" - can you give an example  of a quantified aspect of a job? (e.g. # of deliveries performed during a shift - would this be an example?) 

Socioemotional reward - did you find evidence of workers trying to negotiate good ratings with customers directly? 

interpersonal rewards - is a consequence of lack of interaction with a manager also a lack of ongoing training (or even knowing what a worker needs to perform their work effectively)?

organizational trust - appreciated the discussion about power imbalances. Reinforces how dehumanizing surveillance of these gig workers is

I was surprised to see that your review didn't reveal anything related to (lack of) dignity and experiences of stigma especially given these racialized workers 

Section 5: Health impacts - could you expand a bit more on the mental health impacts of this work? Was there any discussion about the lack of health benefits for these workers given their contractor status? 

Research Needs

Surveillance - is part of the issue also the lack of conceptual clarity regarding what constitutes gig work?

pathways between platform work and health - you address a number of important needs. It would be helpful to also add an equity lens. For example, how does algorithmic management differentially affect the health of women, marginalized workers due to their race/ethnicity, etc. 

No limitations are outlined 
